# Identification of Protein Markers for Chronic Ischemic Heart Disease Through Integrated Analysis of the Human Plasma Proteome and Genome-Wide Association Data

**DOI:** 10.3390/proteomes13040055

**Published:** 2025-11-03

**Authors:** Chunyang Ren, Gan Qiao, Jianping Wu, Yongxiang Lu, Minghua Liu, Chunxiang Zhang

**Affiliations:** 1Department of Pharmacology, School of Pharmacy, Southwest Medical University, Luzhou 646000, China; swmurcy@163.com (C.R.); dqz377977905@swmu.edu.cn (G.Q.); 17865681795@163.com (J.W.); luyx0807@163.com (Y.L.); 2Nucleic Acid Medicine of Luzhou Key Laboratory, Central Nervous System Drug Key Laboratory of Sichuan Province, Southwest Medical University, Luzhou 646000, China; 3Key Laboratory of Medical Electrophysiology, Ministry of Education and Medical Electrophysiological Key Laboratory of Sichuan Province, (Collaborative Innovation Center for Prevention of Cardiovascular Diseases), Institute of Cardiovascular Research, Southwest Medical University, Luzhou 646000, China

**Keywords:** chronic ischemic heart disease, plasma proteome, genome, bidirectional mendelian randomization, drug target

## Abstract

Background: Chronic ischemic heart disease (CIHD) is characterized by persistent myocardial ischemic due to long-term reduced coronary blood flow. In the past, we mainly relied on surgical intervention or drug therapy to alleviate symptoms, but effective targeted treatments were scarce. Proteomics serves as a key tool to identify novel therapeutic targets. Methods: This study performed a bidirectional Mendelian randomization (MR) analysis by integrating genome-wide association study (GWAS) data on CIHD (10,894,596 single-nucleotide polymorphisms) with plasma proteomic data encompassing 4907 proteins. We conducted Kyoto Encyclopedia of Genes and Genomes (KEGG) enrichment analysis to identify pathways linked to candidate protein biomarkers, searched the National Genomics Data Center (NGDC) database for existing evidence of their association with CIHD, and evaluated druggability through multi-dimensional analysis integrating the DSIGDB, ChEMBL, and clinical trial databases. Results: After eliminating the reverse effect, ultimately identifying 28 protein markers, including 16 risk-associated and 12 protective proteins. We also investigated their roles in the pathways related to CIHD. Meanwhile, the search confirmed that five of them were newly discovered protein markers. Ultimately, through evaluation, three priority therapeutic targets (CXCL12, PLAU, CD14) were identified for development. Conclusions: This study identified some biomarkers related to CIHD and analyzed the possible pathways involved. It also provided some new insights into the identification of the target and druggability.

## 1. Introduction

Chronic Ischemic Heart Disease (CIHD) is a cardiac disorder caused by prolonged myocardial ischemia resulting from coronary artery stenosis or occlusion, with major adverse cardiac events strongly correlated with the overall burden of atherosclerosis [1]. The condition includes old myocardial infarction, ischemic cardiomyopathy, coronary atherosclerosis, and other specified or unspecified forms of CIHD, and is primarily characterized by angina pectoris and unstable angina [2]. In 2021, the global prevalence of CIHD among individuals aged 55 years and older reached 182 million, representing 4.2% of this population. CIHD accounted for 9.1 million deaths worldwide, making up 16.2% of all global deaths and ranking as the leading cause of mortality attributable to a single disease. In China, the prevalence of coronary heart disease (including CIHD) in 2021 was 12.3‰ in urban areas and 8.1‰ in rural areas, with a rate of 27.8‰ among individuals over 60 years of age, resulting in more than 16 million affected individuals. These figures underscore the substantial public health burden posed by CIHD both in China and globally [3,4,5]. The clinical management of CIHD primarily involves pharmacological therapies aimed at reducing ischemia and improving prognosis; revascularization procedures such as percutaneous coronary intervention (PCI) and coronary artery bypass grafting (CABG), following comprehensive risk assessment; as well as individualized lifestyle modifications, structured exercise programs, and psychological support to facilitate cardiac rehabilitation [6,7,8,9,10,11]. Notably, novel therapeutic strategies have emerged in recent years. GLP-1 receptor agonists have demonstrated significant improvements in myocardial perfusion reserve and coronary microvascular function through mechanisms including endothelial enhancement and reduction in oxidative stress, particularly in patients with concomitant diabetes mellitus [12]. Additionally, ongoing research is investigating the potential of angiogenic factors and stem cell transplantation to promote collateral circulation development [13]. Given the current limitations in effective therapeutic options and mechanistic interventions, further investigation into the genetic underpinnings and potential therapeutic targets of CIHD remains critically important.

Proteins, the core functional units of life activities, are composed of amino acids linked by peptide bonds and perform diverse biological roles, including catalysis (e.g., enzymes), structural support (e.g., collagen), and signal transduction (e.g., hormone receptors) [14]. Proteomics is a systematic discipline that investigates the entire complement of proteins within an organism. Recent advances in proteomic research have demonstrated its broad applicability in identifying disease-associated protein markers and discovering novel therapeutic targets. For instance, a proteomic analysis of brain tissue from young APOE ε4 carriers revealed aberrant activation of proteins such as STAT3, YES1, and FYN, which was further validated using cellular models, offering potential targets for early intervention in Alzheimer’s disease [15]. Another study integrated proteomic data from 1043 patients across 10 cancer types, leading to the identification of 457 candidate therapeutic targets. This research not only confirmed the known association between TP53 deletion and sensitivity to CHEK1 inhibitors but also uncovered the antineoplastic potential of the antifungal agent Naftifine, thereby providing strong evidence for drug repurposing in oncology [16]. It is worth noting that a single gene can generate multiple protein forms—functionally distinct protein variants—through mechanisms such as alternative splicing and post-translational modifications. By analyzing the complexity of the human proteome, including the various protein isoforms, and identifying disease-specific proteoforms, novel avenues can be opened for disease diagnosis and therapeutic intervention [17].

Mendelian randomisation (MR) is a statistical method that utilises genetic variants as instrumental variables (IVs) to emulate randomised controlled trials. It is grounded in the principle of random gene allocation from parents to offspring during gamete formation and fertilisation. Compared with traditional randomised controlled trials (RCTs), MR minimises confounding by environmental factors and other extraneous variables, thereby enhancing the validity of causal inference. The selection of IVs is based on the exposure factors under investigation, which are subsequently linked with outcome variables for MR analysis. This analytical framework enables the assessment of causal relationships between genetic exposures and outcomes [18]. Numerous studies have applied this methodology to investigate potential associations among various exposure-outcome pairs. MR analysis methods have also been shown to be effective in inferring causal associations, which is an important step in understanding the complex interplay between different factors in medical research [19,20]. Given the wide range of etiologies and numerous influencing factors associated with cardiovascular diseases, it is often challenging to establish direct causal relationships. Bidirectional MR analysis, however, effectively circumvents these confounding factors. Based on this foundation, the present study aims to integrate plasma proteomic and genomic data to systematically identify protein markers that are causally associated with CIHD through bidirectional MR analyses, and to evaluate their potential as therapeutic targets, thereby providing novel insights for mechanistic research and drug development in CIHD. Compared with previous integrative studies, our approach yielded a more robust causal inference by excluding weak instrumental variables and addressing potential reverse causality through bidirectional Mendelian randomization. Furthermore, we identified more promising therapeutic targets for the precision treatment of CIHD via systematic druggability assessment and target prioritization.

## 2. Materials and Methods

### 2.1. Research Design and Datasets

The overall study design is illustrated in Figure 1. Briefly, the initial phase involved conducting a bidirectional MR analysis. A comprehensive panel of 4907 proteins, sourced from a large-scale proteomic study, was used as exposure factors. Additionally, genome-wide association study (GWAS) data on CIHD was utilized as the outcome variable. This analytical strategy was implemented to identify genetic associations between proteins and CIHD. Following this, we conducted an in-depth investigation into biological pathway enrichment and existing evidence and assessed the druggability potential of the identified protein markers.

A large-scale study analysed plasma samples from 35,559 Icelandic individuals, examining the association between 4907 quantified proteins and 373 diseases or traits. This analysis identified 257,490 significant protein-disease associations and highlighted 938 potential drug target genes, whose genetic variants are linked to disease outcomes through their influence on protein levels [21]. The exposure factor dataset used in the present study was derived from this comprehensive research. To assess the technical repeatability of the SomaScan platform for the quantitative determination of 4907 protein abundances, their study also included 228 duplicate tube instances from the same plasma sample. The results showed that the average correlation between the measured values was 0.85 (standard deviation = 0.10). To minimise the potential confounding effects of population stratification, the outcome data for chronic CIHD were obtained from the IEU OPEN GWAS project, which focuses on individuals of European ancestry [22]. This dataset includes a total of 10,894,596 single-nucleotide polymorphisms (SNPs) from 337,199 participants, consisting of 8755 cases and 328,444 controls.

### 2.2. Choosing Genetic Instruments and Multiple Testing Correction

To analyze plasma protein data, protein Quantitative Trait Loci (pQTLs) were selected as IVs, which were subjected to the following screening criteria: (i) SNPs must be located within ±10,000 kb of protein-coding genes. (ii) SNPs must demonstrate a strong association with the abundance of 4907 proteins at genome-wide significance (*p* < 5 × 10^−8^) and retain only common variants (MAF > 0.01). (iii) Linkage disequilibrium (LD) clumping was performed to ensure independence among SNPs (within 10,000 kb, r^2^ < 0.01). (iv) To evaluate the strength of each protein’s abundance as an instrumental variable, the variance explained (R^2^) and F statistic were calculated using standard formulas: R^2^ = 2 × EAF × (1 − EAF) × beta^2^; F = R^2^ × (N − 2)/(1 − R^2^). Following established guidelines, weak IVs with F ≤ 10 were excluded from further analysis [23]. In the reverse MR analysis, where CIHD served as the exposure and plasma proteomics as the outcome, CIHD study data were filtered using the aforementioned criteria genome-wide significance, (iii), and (iv). The resulting outcome dataset included the relevant protein data identified through forward MR analysis [24]. The pQTL screening process was independently conducted twice. Both screenings strictly adhered to identical criteria (±10,000 kb, *p* < 5 × 10^−8^, etc.). Only SNPs that produced consistent results across both screenings were retained as instrumental variables to minimize potential screening bias. This filtering process was conducted using R (Version 4.3.2, with subsequent analyses maintained at the same version), in conjunction with the dplyr (1.1.4) and tidyr (1.1.4) packages.

In view of the fact that this study requires the screening of SNPs associated with protein abundance through cis-pQTLs (involving 4907 proteins, and high-dimensional testing is prone to false positives), multiple testing correction is only performed on the association test between cis-pQTLs and proteins. The Benjamini–Hochberg false discovery rate (FDR) correction is adopted. The specific calculation method is as follows: Arrange the original *p*-values of the associations between cis-pQTLs and all proteins in ascending order and assign a serial number i (i = 1, 2, …). Calculate the corresponding threshold using the formula: FDR critical value = (i × 0.05)/amount. Proteins with significant associations with cis-pQTLs are retained based on the criterion of “original *p*-value ≤ critical value (i.e., FDR < 0.05)”.

### 2.3. Bidirectional MR Analysis and Screening

#### 2.3.1. Forward MR Analysis and Screening

Inverse-variance weighted (IVW) analysis is the primary method in MR, based on the assumption that “instrumental variables influence the outcome exclusively through the exposure factor.” It estimates the causal effect by performing a weighted average, where the weight corresponds to the inverse of the squared standard error of the association between each SNP and the exposure (1/SE^2^). The weighted mode method assumes that “the majority of instrumental variables are free from pleiotropy. The weighted median method assumes that at least 50% of the instrumental variables are not pleiotropic [25]. MR Egger is an extended MR method specifically designed to address scenarios in which the assumptions of traditional IVs are violated, such as when genetic variants exhibit pleiotropy—meaning they influence outcomes through pathways other than the exposure of interest [26]. Simple mode estimator, on the other hand, represents an unweighted mode of the empirical density function of individual causal effect estimates [27]. If the results across all five methods are consistent—showing the same direction of effect and overlapping confidence intervals—it provides strong evidence for the robustness of the causal estimate [28]. To ensure the reliability of the findings, protein markers are subjected to the following screening criteria: (i) *p* < 0.05 in the IVW analysis; (ii) consistent direction of effect (OR > 1 or OR < 1) across all five MR methods; (iii) No evidence of multiplicity was found in the subsequent sensitivity analysis (*p*-value > 0.05). Proteins meeting these criteria are designated as proteins of interest [29].

#### 2.3.2. Reverse MR

Reverse MR analysis is then conducted by switching CIHD as the exposure and the proteins of interest as outcomes, following the same analytical framework. If any of the proteins of interest show significant associations in this reverse analysis, it suggests that these proteins may be genetically influenced by CIHD rather than being independent risk factors. Such proteins are excluded from further consideration. The remaining proteins are defined as candidate protein markers [24].

#### 2.3.3. Sensitivity Analysis

To prevent instrumental variables from influencing the outcome through non-exposure pathways—MR-Egger regression is employed to evaluate horizontal pleiotropy. Cochrane Q test is used to assess heterogeneity among instrumental variables. Additionally, each instrumental variable is sequentially excluded to evaluate the stability of the causal effect estimate, a process known as the leave-one-out analysis, which helps identify potentially influential SNPs [30].

To verify the reliability of causal effect estimation, the entire bidirectional MR analysis was independently repeated three times (using the same dataset and re-running the code). Only proteins showing consistent results and no significant heterogeneity across all three repetitions were retained. These data analyses were performed using R, and R packages including dplyr, gwasglue (0.0.0.9000), TwoSampleMR (0.6.8), MendelianRandomization (0.10.0), and ggplot2 (3.5.2).

### 2.4. KEGG Pathway Analysis

The Kyoto Encyclopedia of Genes and Genomes (KEGG) is an authoritative database that supports the functional analysis of genes and the enrichment of biological pathways. It integrates multidimensional biological data, including metabolic pathways, signal transduction networks, and disease mechanisms [31]. Using the ClusterProfiler package in R, the gene list corresponding to the identified protein markers was uploaded for automated KEGG pathway analysis (organism = ‘hsa’; *p* < 0.05). The list of genes corresponding to the candidate proteins was submitted three times in duplicate. All analyses were performed with the parameters organism = ‘hsa’ and *p* < 0.05. Only pathways that showed significant enrichment (q < 0.05) in all three repetitions were retained, thereby excluding any instances of accidental enrichment. This analysis aimed to identify the signalling pathways in which these markers are involved, thereby revealing potential mechanisms and providing evidence for their role in the regulation of CIHD.

### 2.5. NGDC Search

The National Genomics Data Center (NGDC), as part of the China National Center for Bioinformation (CNCB), primarily offers multi-omics data services to global academic and industrial communities, including large-scale data archiving, integrated analysis, and value-added processing [32]. In our study, we utilized the Cardiovascular Disease Atlas (CVD-A) partition to investigate whether our identified protein markers had been previously associated with either CVD or CIHD (including risk factors, phenotypes, pathological features, etc.), and to assess the potential for co-localization between them and CIHD. After screening the dataset for the European population, we downloaded the relevant data, removed irrelevant variables, and computed the average scores of the remaining factors.

### 2.6. Evaluation of Druggability

To comprehensively evaluate the drug potential and development prospects of the candidate protein markers, a multi-dimensional search and analysis approach was implemented. First, the Drug Signatures Database (DSIGDB) drug/compound–target gene association dataset, was used to perform enrichment analysis by R (*p* < 0.1, adjP < 0.1) on the protein markers [33]. Subsequently, small molecule drug-target interaction data were integrated from the ChEMBL database (version 35) [34]. At the same time, a parallel search for clinical evidence related to these protein markers was conducted using the ClinicalTrials website (www.clinicaltrials.gov, accessed on 10 May 2025). To determine the priority development levels of targets, we constructed a multi-dimensional evaluation framework. This framework comprehensively considers the strength of evidence, pathway participation, and drug accessibility. Each dimension is defined by quantifiable indicators derived from our research findings and validations in public databases:

Strength of Evidence: It is evaluated through three sub-indicators. (1) Genetic Association: It is considered “strong” if the IVW *p* < 0.05 (MR analysis) and there is a clear protective/risk effect (OR ≠ 1); it is “moderate” if only some MR methods support significance. (2) Database Validation: It is “strong” if the NGDC-CVD—A disease score > 0.5 and the colocalization (coloc) score > 0.4 (reflecting a consistent association with CIHD genetics); it is “moderate” if only one score reaches the threshold. (3) Clinical Evidence: It is “strong” if there are ≥5 relevant clinical trials (clinicaltrials.gov); “moderate” if there are 1–4 trials; “weak” if there are none.

Pathway Participation: It focuses on the core pathophysiological pathways of CIHD (such as the NF-κB signaling pathway, complement-coagulation cascade, and fatty acid metabolism). It is considered “high” if it involves ≥3 core pathways or serves as a key regulatory node (e.g., mediating the inflammation-thrombosis interaction); “moderate” if it involves 2 major pathways; “low” if it involves fewer than 2.

Drug Availability: It is evaluated through the following indicators. (1) Drug Enrichment: It is “high” if the number of compounds enriched in DSIGDB is ≥4; “moderate” if it is between 1 and 3. (2) Small-molecule Potential: It is “high” if the number of compounds related to ChEMBL is ≥50; “moderate” if it is between 10 and 49; “low” if it is fewer than 10. (3) Clinical Progress: It is “in progress” if there are approved drugs/clinical candidate drugs; “emerging” if there are only pre-clinical compounds or compounds in the early trial stage.

## 3. Results

### 3.1. Effect of Plasma Proteins on CIHD

Initially, a total of 4907 protein datasets were obtained and subjected to a series of analytical steps. Additionally, Benjamini–Hochberg FDR correction (FDR < 0.05) was performed on the pQTL association results of proteins. This process yielded 5081 SNPs across 880 distinct proteins. Among these instrumental variables, the lowest F-statistic value was 29.72.

Subsequent to conducting forward MR analysis using five directional screening methods, 331 proteins were identified. After applying *p*-value screening, 148 SNPs were obtained from 32 proteins (Figure 2). Among these, 16 proteins were classified as protective factors and 16 as risk factors. When conducting reverse MR analysis on these 32 proteins, four proteins showed evidence of being influenced by exposure data, suggesting potential reverse causation (Figure 3). These four proteins were therefore excluded to avoid bias in the results.

Ultimately, 28 plasma proteins with differential abundance were retained as candidate protein markers, including 12 protective factors (CXCL12, SERPINA9, ADH5, MTHFS, ADH6, PMM2, IGFBP7, SPINK6, ASAH2, SVEP1, FCRLB, ITIH3) and 16 risk factors (IL1RAP, TCN2, ADAMTS13, HP, MICA, CPB2, ELANE, PLAU, ECI2, CD14, IGFALS, PDXK, DNAJB11, PCSK9, LAP3, KLK7) (Figure 4). Regarding the sensitivity analysis, the heterogeneity test using the IVW method showed for protein abundance-associated SNPs Q_*p*-values > 0.1. In the pleiotropy assessment, the MR-Egger intercept was close to zero, with *p*-values > 0.05. Due to the limited number of SNPs, the leave-one-out method is unable to extract meaningful information in this case.

### 3.2. Pathway Analysis

KEGG enrichment analysis revealed that the identified proteins were significantly enriched in several pathways (q < 0.05), including fatty acid degradation (hsa00071, q = 0.0049; 3 genes: *ADH5*, *ADH6*, *ECI2*), the NF-κB signaling pathway (hsa04064, q = 0.0341; 3 genes: *CXCL12*, *PLAU*, *CD14*), tyrosine metabolism (hsa00350, q = 0.0420; 2 genes: *ADH5*, *ADH6*), alcoholic liver disease (hsa04936, q = 0.0420; 3 genes: *ADH5*, *ADH6*, *CD14*), and pyruvate metabolism (hsa00620, q = 0.0445; 2 genes: *ADH5*, *ADH6*) (Figure 5). Transcriptional dysregulation in cancer (hsa05202, q = 0.0570; involving 3 genes: *ELANE*, *PLAU*, and *CD14*), glycolysis/gluconeogenesis (hsa00010, q = 0.0570; involving 2 genes: *ADH5* and *ADH6*), retinol metabolism (hsa00830, q = 0.0570; involving 2 genes: *ADH5* and *ADH6*), drug metabolism-cytochrome P450 (hsa00982, q = 0.0581; involving 2 genes: *ADH5* and *ADH6*), metabolism of xenobiotics by cytochrome P450 (hsa00980, q = 0.0609; involving 2 genes: *ADH5* and *ADH6*), and complement and coagulation cascades (hsa04610, q = 0.0680; involving 2 genes: *CPB2* and *PLAU*) were identified with q values ranging from 0.05 to 0.1.

Functional annotations further highlight the roles of key proteins: ADH5 and ADH6, appear repeatedly across eight pathways and belong to the alcohol dehydrogenase family. These enzymes not only metabolise ethanol but also regulate redox homeostasis of fatty acids and aldehydes. Abnormal abundance of these proteins can lead to lipid peroxidation and disturbances in energy metabolism, directly affecting myocardial cell survival. PLAU, present in three pathways, regulates fibrinolysis, inflammation, and cell migration, potentially contributing to thrombus formation, plaque stability, and myocardial repair in CIHD. CD14, involved in two disease-related pathways, acts as an endotoxin receptor that mediates inflammatory signal transduction and is strongly associated with inflammatory mechanisms underlying atherosclerosis.

### 3.3. Existing Evidence

A systematic search of protein markers has provided evidence that 26 markers are associated with cardiovascular diseases (A comprehensive correlation table can be found in the attachment). Notably, among these 26 markers, 23 exhibit a strong correlation with coronary artery disease CIHD in terms of associated diseases or traits, and 19 demonstrate a co-localization relationship. The key contributing factors include lipid profiles, blood pressure levels, blood glucose levels, and pathological features of CIHD. The association scores for these markers were collected and averaged, with the results presented in Figure 6. Currently, there is insufficient data to establish a clear association for ASAH2 and SERPINA9. Furthermore, ECI2, KLK7, and SPINK6 appear to be linked to other forms of cardiovascular disease rather than CIHD.

### 3.4. Druggability

Using the hypergeometric test model based on the DSIGDB database, 10 drugs or compounds were found to be significantly enriched for 11 out of the 28 protein markers (Figure 7). The remaining 17 targets showed no significant enrichment. The identified compounds include small-molecule agents (e.g., phorbol ester, gemcitabine), natural products (withaferin A), and toxicants (hexachloroethane), highlighting the diverse functional roles of these targets in biological processes such as inflammation regulation, thrombosis, and immune surveillance.

Among these, PLAU was enriched by six drugs (phorbol ester, withaferin A, 1,9-pyrazoloanthrone, etc.), with phorbol ester (*p* = 0.000377) shown to upregulate PLAU abundance via activation of protein kinase C (PKC). Notably, the enrichment of PLAU by heavy metal salts (e.g., sodium dichromate; *p* = 0.002540) suggests that oxidative stress may modulate coagulation function through PLAU. CXCL12 was enriched by four compounds (terbinafine, gemcitabine, withaferin A, and glycoprotein). Among them, gemcitabine (*p* = 0.001820) inhibits tumor cell migration by targeting the CXCL12/CXCR4 axis, potentially suppressing monocyte infiltration into atherosclerotic plaques through a similar mechanism. Glycoprotein enrichment (*p* = 0.001935) indicates that glycosylation modifications of CXCL12 may influence its receptor-binding affinity, suggesting potential structural targets for drug design optimisation. ADAMTS13 was enriched by six compounds, including hexachloroethane and gemcitabine. Hexachloroethane, a halogenated hydrocarbon (*p* = 0.001964), exhibited an enrichment profile highly consistent with ADAMTS13’s known role in thrombotic thrombocytopenic purpura (TTP).

A comprehensive search of the ChEMBL database (Version 35) and clinical trial registries was conducted for all 28 protein markers. Based on the findings, the proteins were categorised into three developmental stages: Approved Drugs and Clinical Candidates (n = 4), those currently in clinical development (n = 8), and those requiring further preclinical investigation (n = 16) (Figure 8).

EVOLOCUMAB and ALIROCUMAB are well-established pharmaceutical drugs that have been approved and on the market for over a decade. Several other candidates, including BOCOCIZUMAB, TAFOLECIMAB, LERODALCIBEP, and ONGERICIMAB, are currently in Phase III clinical trials, while others such as FROVOCIMAB remain in Phase II development. The primary therapeutic indications for these drugs include hypercholesterolemia, coronary artery disease, myocardial infarction, and hyperlipidaemia—conditions collectively classified under cardiovascular diseases—all of which target and inhibit PCSK9 activity. Two IL1RAP-targeting drugs, SPESOLIMAB and IMSIDOLIMAB, have already received regulatory approval for the treatment of inflammatory conditions such as psoriasis, ichthyosis, and Crohn’s disease. N6022 and CAVOSONSTAT are small-molecule inhibitors of alcohol dehydrogenase (ADH), specifically targeting the ADH5 isoform. These compounds are being investigated for the treatment of chronic obstructive pulmonary disease, cystic fibrosis, and asthma. Additionally, OLAPTESED PEGOL, a CXCRL14 antagonist, is currently undergoing early-phase clinical trials for the treatment of pancreatic neoplasms and glioblastoma.

CD14, PMM2, HP, MICA, ADAMTS13, ELANE, PLAU, and IGFBP7—eight proteins in total—are currently the subjects of ongoing clinical development trials. Notably, one trial is evaluating the efficacy of individua lised capreomycin monotherapy in adults with acquired TTP, while another is investigating ALT 2074 in type 2 diabetes patients carrying the 2-2 HP genotype who also have coronary artery disease. Additional studies are assessing oral eptifibatide therapy for children with PMM2-CDG, as well as multiple trials examining GLM101 across various outcome measures in PMM2-CDG patients. Regarding CD14, a preliminary study is currently recruiting participants to evaluate the use of an anti-CD14 monoclonal antibody in the treatment of ST-segment elevation myocardial infarction. Another study is utilizing soluble CD14 as a surrogate biomarker to assess the anti-inflammatory effects of apixaban in reducing diabetic macular edema (DME). Although clinical trial data exist for the remaining four proteins, their direct involvement in CIHD has not yet been clearly established.

Currently, the association between 16 types of proteins, including KLK7 and PDXK, and any existing drugs remains undetermined, with no relevant clinical evidence available to date. These proteins may serve as potential targets for the development of novel therapeutic agents in the future.

Based on the above results, the three protein markers, CXCL12, PLAU and CD14, demonstrated strong druggability (Table 1) (More information can be found in Appendix B Table A1).

## 4. Discussion

This study integrated genomic data of 4907 plasma proteins and CIHD, and identified 28 protein markers (12 protective factors and 16 risk factors) with significant genetic associations with CIHD via bidirectional MR analysis. Kyoto Encyclopedia of Genes and Genomes (KEGG) enrichment analysis revealed that metabolic pathways (e.g., fatty acid degradation, glycolysis/gluconeogenesis) and inflammatory pathways (e.g., NF-κB signaling pathway, complement and coagulation cascades) are functionally interconnected through key genes such as ADH5/ADH6 and PLAU [35]. Specifically, abnormal fatty acid degradation mediated by aberrant abundance of ADH5/ADH6 may trigger energy metabolism dysfunction in cardiomyocytes—consistent with previous findings that metabolic enzyme dysregulation can act as a trigger for oxidative stress in ischemic myocardium [36]—which further activates the NF-κB pathway via the accumulation of reactive oxygen species (ROS); meanwhile, the inflammatory response induced by PLAU/CD14 can suppress the activity of mitochondrial β-oxidation enzymes, forming a self-perpetuating cycle of “metabolic disturbance→oxidative stress→inflammatory activation→progressive metabolic impairment” [37]. A search in the National Genomics Data Center (NGDC) showed that among these 28 markers, 23 have been previously associated with coronary artery disease (CIHD), and the remaining 5 (ASAH2, SERPINA9, ECI2, KLK7, SPINK6) are newly identified genetic markers linked to CIHD. In addition, Drug Signatures Database (DSIGDB) analysis identified 10 drugs or compounds significantly enriched for 11 of these CIHD-associated protein markers, and further investigation revealed 4 proteins targeted by approved drugs or clinical candidates, as well as 8 proteins with therapeutic potential in clinical development [38,39].

The core mechanism of CIHD lies in the imbalance between myocardial oxygen supply and demand, which is attributed to multiple factors including fixed plaques or dynamic spasms in epicardial vessels or microvessels, platelet microembolism, endothelial dysfunction-induced impaired vascular dilation, increased vascular smooth muscle reactivity, myocardial hypertrophy, and microcirculatory dysfunction [40]. Traditionally, treatment strategies have focused on regulating oxygen supply and demand through nitrates, beta-blockers, and calcium channel blockers; emerging approaches have explored microcirculatory dilation, endothelial protection, vascular remodeling regulation, and metabolic optimization [41]. However, current treatments still have limitations, and there is an urgent need to develop new therapeutic targets or drugs. The “metabolism-inflammation” cross-regulatory network identified in this study is highly consistent with the core pathophysiological mechanisms of CIHD, providing a potential direction for the development of novel therapeutic strategies—such as targeted enhancement of protective pathways, precision inhibition of risk factors, and combination therapy targeting the “metabolism-inflammation” crosstalk.

In this “metabolism-inflammation” regulatory network, the role of key proteins and pathways is particularly notable: ADH5/ADH6, which are involved in fatty acid degradation, are closely associated with energy metabolism reprogramming in ischemic myocardium, and their abnormal abundance directly contributes to metabolic disturbances; CXCL12 mediates leukocyte recruitment through the CXCR4/CXCL12 axis, a process that plays a central role in atherosclerotic inflammation; PLAU is enriched in the complement and coagulation cascade, supporting its dual involvement in inflammatory and thrombotic processes, and its gene polymorphisms have been linked to thrombophilia; CD14, as an endotoxin receptor, mediates inflammatory signal transduction, and elevated CD14 levels are associated with increased cholesterol levels and higher risks of myocardial infarction and cardiovascular mortality [42,43]. Additionally, members of the transforming growth factor-beta (TGF-β) superfamily (INHBA, INHBB, NOG) occupy central positions in the protein interaction network, reflecting the dual role of the TGF-β pathway (regulating myocardial fibrosis and vascular plaque stability) at different disease stages [44,45,46]. From a therapeutic translation perspective, these targets can be developed in tiers: PCSK9, an established therapeutic target with clinical applications, serves as a benchmark [47]; priority should be given to proteins with strong therapeutic potential such as CXCL12, PLAU, and CD14; next are targets with evidence from drug development or clinical trials (e.g., IL1RAP, ADH5, PMM2, HP, MICA, ADAMTS13, ELANE, IGFBP7) or enrichment across multiple pathways (e.g., ADH5, ADH6); the remaining unexplored targets may be promising directions for future research. It is worth noting that potential side effects of drugs (e.g., the vasotoxicity of gemcitabine [48]) need to be carefully considered during clinical application.

Nevertheless, the present study has several limitations that warrant consideration (More information can be found in Appendix B Table A2). First, the protein markers were identified based on data derived from European populations—including Icelandic plasma proteome data and European CIHD GWAS data—and their ethnic specificity may constrain generalizability to other racial or ethnic groups. For example, the concentration of HP protein-related inflammatory complexes in South Asian males is significantly lower than that in white males (median: 2.5 vs. 3.2 ng/L, *p* < 0.001) [49]. The difference may stem from genetic background (such as linkage disequilibrium patterns) and environmental factors (such as diet, metabolic habits) that are specific to different races [50]. Second, MR analysis depends on the strength of the association between genetic variants and protein abundance levels. It has been shown that some low-abundance proteins may be overlooked due to limited availability of pQTL data [51]. Despite the technical repeatability of the SomaScan platform (average correlation = 0.85 for duplicate plasma samples) and its utility in high-throughput plasma proteome profiling, this study is constrained by inherent limitations of aptamer-based affinity methods—particularly inconsistencies with non-affinity proteomic technologies (e.g., liquid chromatography-tandem mass spectrometry, LC-MS/MS; antibody-based Olink) [52]. Therefore, it is essential to employ complementary technologies (such as targeted mass spectrometry) to validate key protein biomarkers in the future, particularly when applying the research findings to clinical development. Meanwhile, due to the influence of the number of SNPs, the reliability of MR results cannot be adequately assessed using the leave-one-out method. Therefore, future studies may benefit from incorporating additional datasets to further validate these findings. The comorbidities, such as diabetes and hypertension, as well as the medication use of the research subjects, were not adequately investigated, despite their potential to directly influence plasma protein abundance levels [53].

The present study can be further advanced through some key research directions. Firstly, single-cell proteomics may be utilized to investigate the abundance profiles of the 28 identified proteins across specific cell types, such as cardiomyocytes and endothelial cells, with the objective of elucidating their cell-specific functional mechanisms and providing a foundation for cell-specific targeted therapies. Secondly, the establishment of animal models is imperative to validate the biological roles of key proteins, including those of the following genes: CXCL12, PLAU, CD14, and ADH5. This validation should be conducted through gene knockout or knockdown experiments, which are designed to assess their impact on post-ischemic myocardial remodelling and the associated molecular pathways. Thirdly, multi centre clinical trials are indicated to evaluate the synergistic therapeutic effects of targeted combination strategies, such as PCSK9 + CXCL12, with a particular focus on differential treatment outcomes in patients exhibiting metabolic abnormalities, including diabetes and obesity. Fourthly, the repurposing of pharmaceuticals can be pursued by evaluating the existing non-cardiovascular therapeutic agents in relation to the identified protein markers. For instance, it is important to determine whether gemcitabine directly inhibits the abundance or activity of CXCL12 through in vitro and in vivo experimental models, and to assess the resulting effects on cardiomyocytes, vascular endothelial cells, and immune cell populations. Furthermore, these findings require further validation in Asian and African populations: On one hand, validate the consistency of the causal chain of pQTLs-proteins-CIHD across different ethnic groups using large—sample plasma proteome and GWAS data; on the other hand, optimize the ethnic adaptability of biomarkers by integrating ethnic-specific genetic varia-tions, ultimately enhancing the global universality of the research findings.

## 5. Conclusions

In this study, we integrated proteomic and genomic data using bidirectional Mendelian randomization to identify 28 protein markers with genetic associations to coronary ischemic heart disease (CIHD), as inferred from plasma protein abundance. Among these, 12 were protective factors and 16 were risk factors—five of which (ASAH2, SERPINA9, ECI2, KLK7, SPINK6) represent novel discoveries. We further uncovered a “metabolic-inflammatory” regulatory network underlying CIHD pathogenesis, in which key proteins—including ADH5 and ADH6 involved in fatty acid degradation, and PLAU and CD14 implicated in NF-κB signaling—mediate crosstalk between metabolic dysregulation and inflammatory activation, thereby providing a mechanistic foundation for targeted therapeutic interventions. Notably, three prioritized proteins—CXCL12, PLAU, and CD14—demonstrate strong target potential. Furthermore, the five newly identified markers expand the repertoire of candidates for preclinical exploration, while established markers such as PCSK9—with clinically approved inhibitors—support the robustness of our analytical framework.

For clinical translation, these findings offer dual utility. In risk stratification, proteins including CD14—associated with myocardial infarction risk—and HP—linked to lipid metabolism—could be incorporated into a multi-marker panel to improve identification of high-risk individuals. In drug development, the “metabolic-inflammatory” axis enables precision targeting: for instance, enhancing ADH5/ADH6 activity may ameliorate myocardial energy deficits, while inhibiting PLAU could disrupt pathological thrombosis-inflammation feedback loops. Collectively, this work bridges molecular insights with clinical applications, offering actionable strategies for biomarker-guided risk assessment and rational design of novel therapeutics for CIHD.

## Figures and Tables

**Figure 1 proteomes-13-00055-f001:**
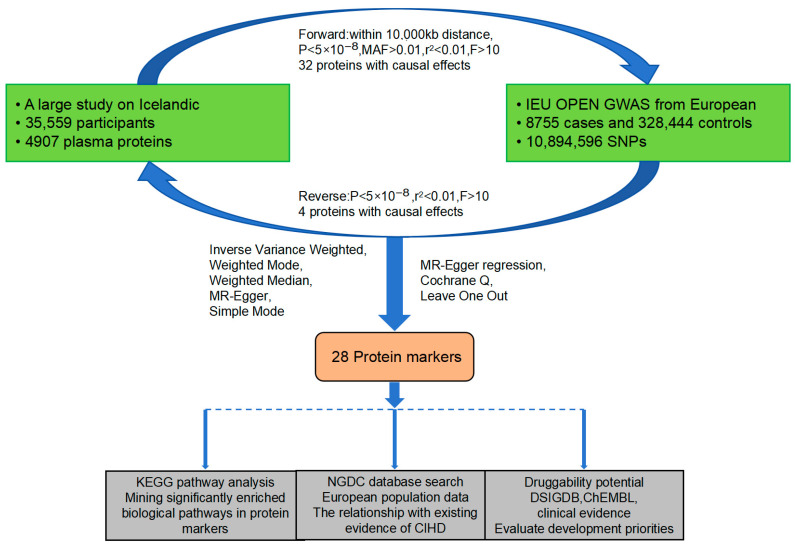
Flowchart of the research design. The core objective is to eliminate reverse causal interference through bidirectional MR analysis, while integrating pathway enrichment, database validation, and drugability assessment to ensure that the selected protein markers possess both “causal association” and “clinical translational value”. Some steps (such as instrumental variable selection and MR analysis) have been repeatedly verified to reduce bias.

**Figure 2 proteomes-13-00055-f002:**
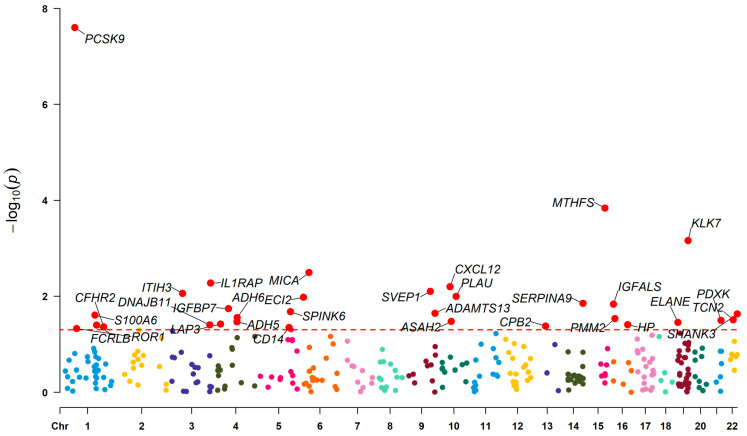
Manhattan plot for association screening based on the directional consistency results of MR analysis and five methods (the red horizontal line represents the significance threshold of *p* = 0.05, and the genes highlighted in red indicate significant associations). The x-axis (Chr) represents the chromosome numbers of the 22 human autosomes (each in a distinct color), indicating the genomic positions of the protein-coding genes(). The y-axis (−log_10_
*p*) indicates the statistical significance of the association between each protein and CIHD, with higher values reflecting stronger evidence of association. (Note: This figure is based on results from three independent MR analyses. All significantly associated proteins highlighted in red (*p* < 0.05) fulfilled the criteria of consistent effect direction across all five methods and no significant pleiotropy in each of the three repetitions. The observed results demonstrate stability and reliability).

**Figure 3 proteomes-13-00055-f003:**
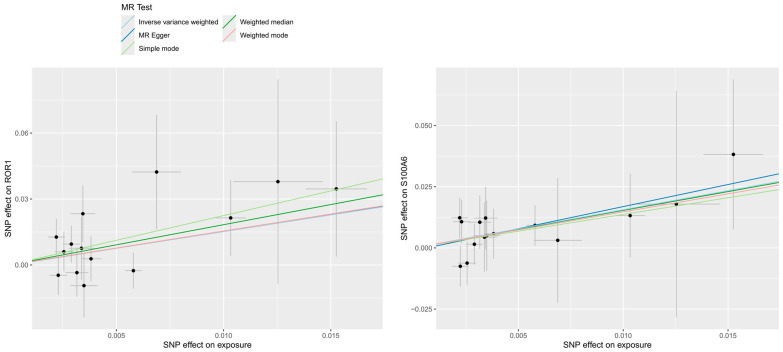
Reverse MR scatter plot: This figure presents the reverse MR analysis for four proteins (ROR1, S100A6, SHANK3, CFHR2), with CIHD as the exposure and each protein as the outcome. The analysis includes four subplots, each corresponding to one of the proteins in sequence. The x-axis represents the genetic effect of SNPs on CIHD (β value); the y-axis represents the genetic effect of SNPs on the respective protein (β value). The lines are colored to distinguish results from five MR methods (e.g., blue: IVW method; red: weighted median method).

**Figure 4 proteomes-13-00055-f004:**
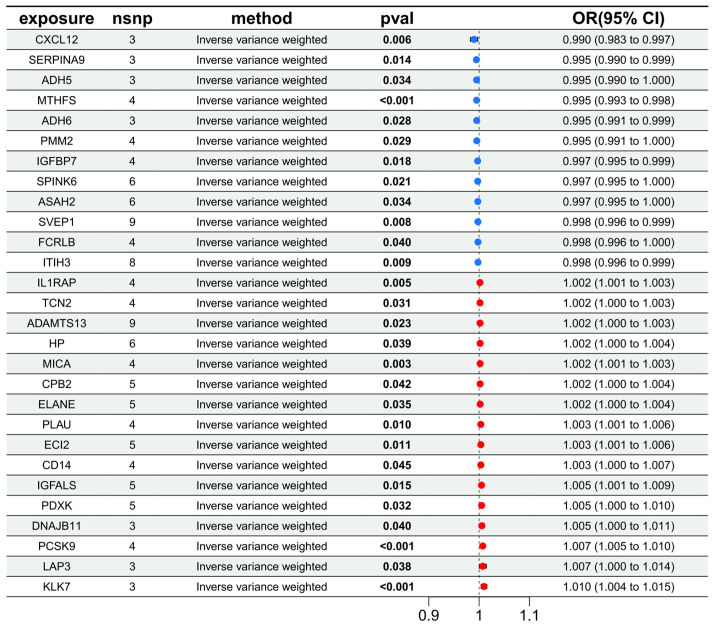
Forest plot of the association between exposure and outcome using the IVW method for protein markers (showing the number of SNPs, *p* values, OR value, and 95% confidence intervals for each exposure factor). Horizontal axis (OR): Ratio of causal effect (OR < 1 = protective factor, OR > 1 = risk factor); Black dotted line: Zero effect line (OR = 1) (the blue origin represents a protective factor, the red dot represents a risk factor). (Note: The OR values and 95% confidence intervals in the figure represent the mean estimates from three independent IVW analyses; the number of SNPs for each exposure factor corresponds to the intersection of twice pQTL screenings).

**Figure 5 proteomes-13-00055-f005:**
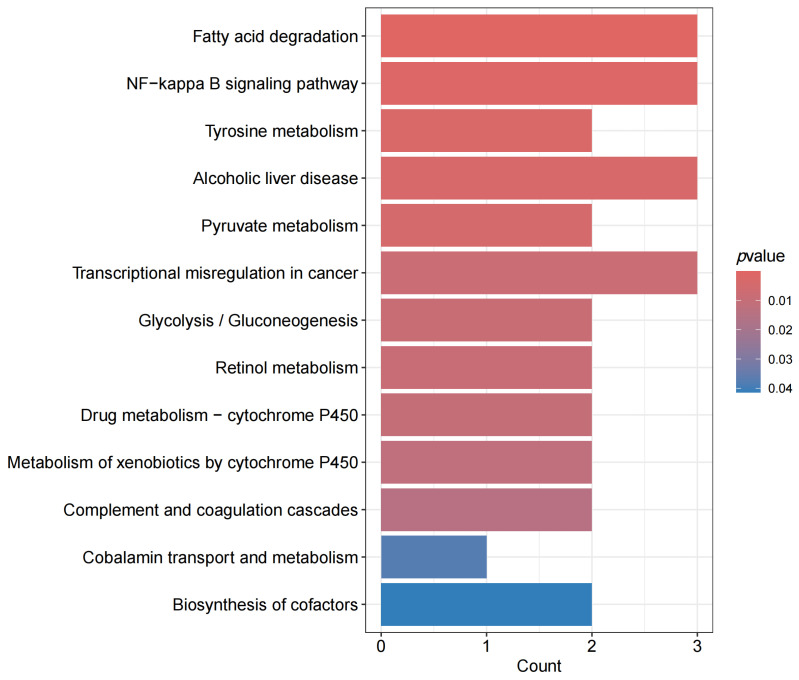
Barplot of KEGG enrichment analysis results. Y-axis: Name of the KEGG pathway; X-axis: Number of candidate proteins contained in the pathway; The length of the column chart is based on the quantity, while the color depth is based on the *p* value. (Note: All the pathways marked in this figure were obtained through three independent repeated KEGG enrichment analyses.).

**Figure 6 proteomes-13-00055-f006:**
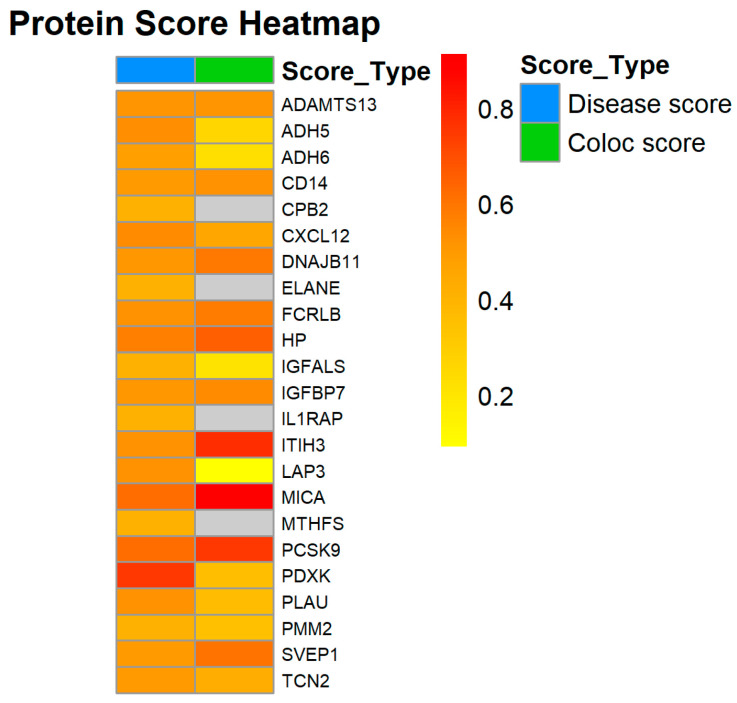
Heatmap of NGDC retrieval results (displays the associations between proteins and diseases/traits, as well as the distribution of average colocalization scores; gray represents score = 0). Vertical axis (Protein): Lists the names of all 23 associated protein markers (such as ADH5, CXCL12, CD14, etc.); Horizontal axis (Score Type): Includes two types of scores—① “Disease score (disease score)”: Reflects the known association strength of the protein with CIHD and related traits (such as blood lipids, blood pressure), calculated based on the cardiovascular disease map (CVD-A) in the NGDC database for the European population; ② “Coloc score (co-localization score)”: Reflects the genetic co-localization probability of the gene locus corresponding to the protein and the susceptibility loci of CIHD.

**Figure 7 proteomes-13-00055-f007:**
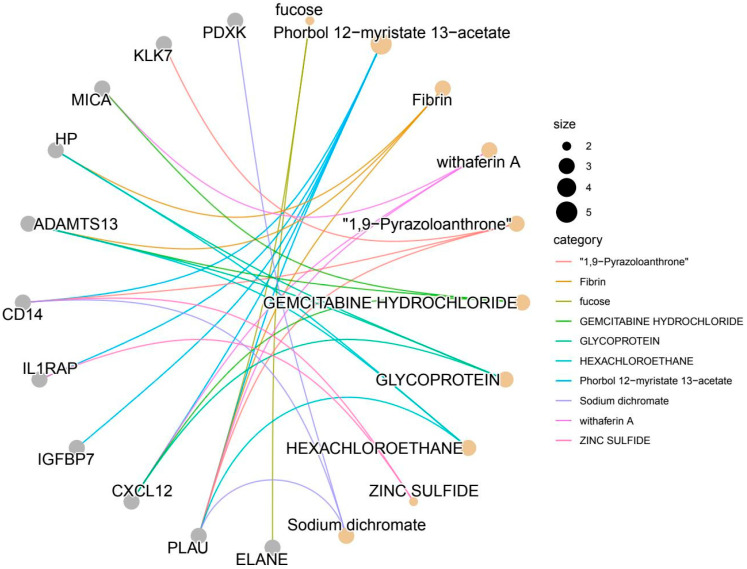
Cnetplot of protein-drug enrichment relationships (showing the relationships between 11 protein markers and 10 enriched drugs). Node type: Blue nodes: Represent candidate protein markers; Orange nodes: Represent enriched drugs/compounds selected from the DSIGDB database; Connection meaning: The black lines connecting proteins and drugs indicate a significant enrichment association between the drug and the protein (hypergeometric test *p* < 0.1, corrected *p* < 0.1); Key example: PLAU (urokinase-type plasminogen activator) is associated with 6 drugs (the most extensive association), while CXCL12 is associated with 4 drugs, suggesting that these proteins have a high potential for drug targeting.

**Figure 8 proteomes-13-00055-f008:**
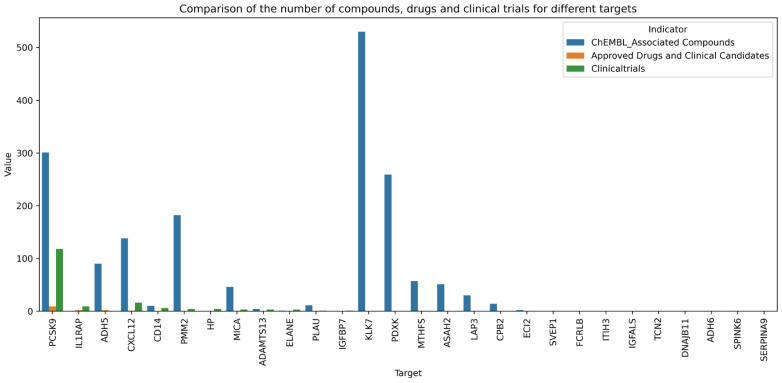
Charts of the retrieval results from databases for protein markers/targets (showing the number of related compounds of 28 protein markers, the approved drugs or the drugs currently in clinical trials, as well as the corresponding clinical trials). Horizontal axis (Target): Lists all 28 candidate protein biomarker names; Vertical axis (Value): Corresponds to the quantity of three types of data—“ChEMBL_Associated Compounds”: The number of small molecule compounds that bind to proteins in the ChEMBL database (version 35); “Approved Drugs and Clinical Candidates”: The number of approved drugs or clinical-stage candidate drugs; “Clinicaltrials”: The number of clinical trials on the ClinicalTrials website that target or use this protein as a biomarker.

**Table 1 proteomes-13-00055-t001:** Comprehensive evaluation table for the key targets of CIHD.

Protein Markers	Effect	Strength of Evidence	Pathway Involvement	Drugability	Reasons
CXCL12	Protective factor	Genetic association: IVW *p* = 0.006, OR = 0.990 (significantly protective)	Core pathway: NF-κB signaling (regulating inflammation), CXCR4/CXCL12 axis (atherosclerosis)	Drug enrichment: 4 compounds (such as gemcitabine) are enriched (DSIGDB)	It exhibits the strongest genetic association (*p* = 0.006), the most comprehensive clinical evidence (16 trials), remarkable druggability (138 compounds + candidate drugs), and directly regulates the core inflammatory axis of CIHD.
		Database support: NGDC disease score = 0.540, coloc = 0.461 (both high)	Number of pathways: 2 core pathways	Number of compounds: 138 (ChEMBL)	
		Clinical evidence: 16 related trials (clinicaltrials.gov)	Function: Inhibiting monocyte infiltration (key anti-inflammatory step)	Clinical progress: OLAPTESED PEGOL (CXCR4 antagonist, early trial)	
		Overall rating: High	Overall rating: Medium	Overall rating: High	
PLAU	Risk factor	Genetic association: IVW *p* = 0.010, OR = 1.003 (significant risk)	Core pathway: NF-κB signaling, complement—coagulation cascade, transcriptional dysregulation (cancer/cardiovascular crossover)	Drug enrichment: 6 compounds (such as phorbol ester) are enriched (DSIGDB, up to)	This pathway exhibits the highest level of participation (covering inflammation, coagulation, and transcriptional regulation) and the greatest degree of drug enrichment (6 types of drugs). However, there is limited clinical evidence (only 1 case). Therefore, it is necessary to promote its clinical translation.
		Database support: NGDC disease score = 0.528, coloc = 0.364 (single high)	Number of pathways: 3 core pathways	Number of compounds: 11 (ChEMBL)	
		Clinical evidence: 1 relevant trial	Function: Regulates thrombosis + plaque stability (dual core step)	Clinical progress: No marketed drug, but multi-drug regulatory potential (such as fuprostone)	
		Overall rating: Medium	Overall rating: High	Overall rating: Medium	

Note: rating (high/medium/low).

## Data Availability

The original data presented in the research report are provided in the article and its Appendix A. For any additional information, please do not hesitate to contact the corresponding author. In the MR analysis, protein data can be accessed from deCODE’s database (www.decode.com/summarydata, accessed on 5 April 2025), whereas CIHD data is sourced from the IEU Open GWAS platform (https://opengwas.io, accessed on 8 April 2025). Other data can be obtained from publicly accessible databases such as NGDC, DSIGDB, ChEMBL, and the ClinicalTrials website. The supporting data, custom code, and analysis scripts for this study will be made available through an open GitHub repository. Access codes and direct repository links will be provided upon publication.

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
