# Peer review of "Identification of Protein Markers for Chronic Ischemic Heart Disease Through Integrated Analysis of the Human Plasma Proteome and Genome-Wide Association Data"

_proteomes, 2025, doi:10.3390/proteomes13040055_

Round 1

Reviewer 1 Report

Comments and Suggestions for Authors

The manuscript is well structured, with a coherent and logical organization that facilitates the reader’s understanding. The methodology is thoroughly described and appropriately presented, ensuring both transparency and replicability. The results are clearly reported and systematically organized, allowing for a sound interpretation of the findings. The discussion section effectively emphasizes the relevance of the results while also acknowledging the study’s limitations in a balanced and critical manner. Furthermore, the references cited are appropriate and reflect the research in the field.

In summary, the manuscript demonstrates scientific merit and makes a valuable contribution to the literature. I recommend it for publication, contingent upon minor revisions to address small textual inaccuracies.

The title have a missing letter "Dentification of protein markers for Chronic Ischemic Heart 2 Disease through integrated analysis of the human plasma pro-3 teome and genome-wide association data" and also subtitle "2.3. Idirectional MR analysis and screening".

Author Response

Commment:The title have a missing letter "Dentification of protein markers for Chronic Ischemic Heart 2 Disease through integrated analysis of the human plasma pro-3 teome and genome-wide association data" and also subtitle "2.3. Idirectional MR analysis and screening".

Response: We completely agree with the mistakes you pointed out.We have changed the title to " Identification of protein markers for Chronic Ischemic Heart Disease through integrated analysis of the human plasma proteome and genome-wide association data " and the subtitle to " 2.3. Bidirectional MR analysis and screening ".Thank you for your comments and suggestions. 

Reviewer 2 Report

Comments and Suggestions for Authors

The article "Dentification of protein markers for Chronic Ischemic Heart Disease through integrated analysis of the human plasma proteome and genome-wide association data" presents a compelling and well-structured study that uses a sophisticated Mendelian randomization approach to identify and validate protein markers for chronic ischemic heart disease. The authors used a robust research design, integrating large-scale proteomic and genomic datasets to invesigate causal relationships. The significance of the findings is based on valuable insights for future chronic ischemic heart disease research and drug development they offer.

The use of bidirectional Mendelian randomization is a key strength, as it helps minimizing confounding factors and reverse causality. Furthermore, the use of multiple MR methods adds to significance and ensures the robustness and reliability of the results. The study also effectively combines data from multiple sources which provides a thorough and well-supported analysis. Finally, it identifies five new protein markers not previously associated with chronic ischemic heart disease, which is a significant contribution to the field.  

The suggestions for authors are as follows:

  • In the Introduction section, the authors could consider adding a sentence or two to bridge the gap between the general background on chronic ischemic heart disease and the specific methodology of Mendelian randomization. It could be, for example, an explanation why a method like Mendelian randomization is particularly necessary for a complex disease like chronic ischemic heart disease, where direct causal links are difficult to establish.
  • Lines 253-254 – It is stated that due to the limited number of SNPs, the leave-one-out method was unable to extract meaningful information in this case. A discussion of how this limitation could be addressed in future studies (e.g., using larger datasets) would be beneficial in the limitations paragraph of the Discussion section.
  • In the Discussion section, the authors could elaborate on the biological significance of the identified protective and risk factors and how do these findings provide new therapeutic approaches.
  • Finally, in the limitations paragraph, the reliance on datasets from European populations should be added, since the same may limit the generalizability of the findings to other ethnic groups. 

Author Response

Comments 1:In the Introduction section, the authors could consider adding a sentence or two to bridge the gap between the general background on chronic ischemic heart disease and the specific methodology of Mendelian randomization. It could be, for example, an explanation why a method like Mendelian randomization is particularly necessary for a complex disease like chronic ischemic heart disease, where direct causal links are difficult to establish.

Response 1:Thank you for your comment. We believe this opinion is both valid and necessary. For this, we added the following words on line 103:”Given the wide range of etiologies and numerous influencing factors associated with cardiovascular diseases, it is often challenging to establish direct causal relationships. Bidirectional MR analysis, however, effectively circumvents these confounding factors”.

Comments 2:Lines 253-254-It is stated that due to the limited number of SNPs, the leave-one-out method was unable to extract meaningful information in this case. A discussion of how this limitation could be addressed in future studies (e.g., using larger datasets) would be beneficial in the limitations paragraph of the Discussion section.

Response 2:The suggestion has been adopted by us, and we have added the following to the "limitation" section of the discussion:”Meanwhile, due to the influence of the number of SNPs, the reliability of MR results cannot be adequately assessed using the leave-one-out method. Therefore, future studies may benefit from incorporating additional datasets to further validate these findings.”

Comments 3:In the Discussion section, the authors could elaborate on the biological significance of the identified protective and risk factors and how do these findings provide new therapeutic approaches.

Response 3:This suggestion has provided us with profound inspiration. So we had more discussions before getting to the "finally" in the first paragraph of the discussion section:”Among the 12 protective factors, three key proteins—CXCL12, ADH5, and ADH6—exert critical roles in maintaining myocardial homeostasis by targeting CIHD’s core pathological cascades (metabolic imbalance and inflammatory activation).The 16 risk factors primarily drive CIHD progression by amplifying lipid dysregulation, inflammatory activation, and thrombotic formation—PCSK9, CD14, and PLAU are the most clinically relevant due to their well-supported pathogenic roles and druggability. These biological insights directly inform three novel therapeutic strategies to address CIHD’s unmet needs: targeted enhancement of protective pathways; precision inhibition of risk factors; and combination therapy targeting the "metabolism-inflammation" crosstalk. ”

Comments 4:Finally, in the limitations paragraph, the reliance on datasets from European populations should be added, since the same may limit the generalizability of the findings to other ethnic groups.

Response 4:It was due to our lack of thorough consideration that this part was not fully discussed. After considering this suggestion, we will modify the second sentence of paragraph A as follows: “First, the protein markers were identified based on data derived from European populations—including Icelandic plasma proteome data and European CIHD GWAS data—and their ethnic specificity may constrain generalizability to other racial or ethnic groups.”

Due to the editing requirements of the manuscript, we are unable to provide you with consistent line numbers. You can find more detailed modification information in the subsequent updated manuscripts you upload.

Reviewer 3 Report

Comments and Suggestions for Authors

This manuscript presents a meticulously organized and thorough investigation that amalgamates extensive plasma proteomic data with genome-wide association studies, employing bidirectional Mendelian Randomization to pinpoint possible protein biomarkers and therapeutic targets for chronic ischemic heart disease (CIHD). The scientific justification is compelling, and the analytical approach encompassing MR analysis, KEGG pathway enrichment, NGDC validation, and druggability evaluation is solid and suitable for the outlined aims. The manuscript is clear and comprehensive, contains detailed methodological descriptions and demonstrates meticulous discussion correlating molecular findings with CIHD pathophysiology and therapeutic potential. Clarity, brevity, methodological explanation, as well as presentation of figures, in some areas could be enriched for enhancing the overall impact of the paper.

1. It is relevant and innovative combining plasma proteomics with bidirectional MR for CIHD. Nevertheless, the authors should more directly point out the innovation of their study in contrast with past proteomic GWAS integration studies (e.g. Ferkingstad et al., Nature Genetics 2021) in the Introduction.

2. The Abstract is fairly compact. Think about making some sentences shorter and quantifying important findings, for example, number of proteins investigated, number of markers detected, and significant pathways, so the abstract is more readable for the wider audience.

3. Graphics, such as the Manhattan plot, the forest plot, and the KEGG barplot, are indispensable; however, the legends are not detailed enough. It is important to include exhaustive descriptions of the axes, thresholds of significance, and key color codes. Also, Figures 2-5 would benefit from higher resolution as well as clearer presentation for clearer reading.

4. Although the flowchart is useful, it could also be improved by the addition of data sources, sample numbers, and crucial analytical protocols, for example, the number of proteins screened at each stage, in order to help the reader negotiate the complex workflow.

5. It details the MR methods, but the presentation order got mixed up. It might help to reorder according to the actual pipeline: choosing the instruments, forward MR, reverse MR, and testing for sensitivities. It will also help to provide a schematic summary table for the MR methods and assumptions.

6. Explain how multiple testing correction was applied during the course of the MR analyses (e.g., Bonferroni, FDR). It's relevant given the number of proteins screened.

7. The Discussion cursorily states that the data set consists of European cases only. That is a severe limitation. Authors should elaborate on possible ethnic variation in protein QTLs for CIHD and genetics and how future validation in non-European populations could rectify it.

8. The Discussion nicely connects metabolic as well as inflammatory pathways, but the explanation at times is repetitive, particularly for ADH5 and ADH6 in metabolism. Think about simplification in order to highlight the core metabolism–inflammation–regulating axis as well as how it guides therapeutic targeting.

9. Comprehensive druggability analysis. Nevertheless, criteria for prioritizing therapeutic targets for treatment (CXCL12, PLAU, CD14) might be specified more clearly. It would greatly assist the ranking of candidate targets by evidence strength, involvement in pathway(s), and druggability using a table summarizing the ranking.

10. Their shortcomings are listed, but could have been more organized as well as complete. For example:
Confounding by comorbidities, Single ancestry limitation, Potential pQTL detection bias, and Lack of experimental validation. It would make them more readable to list them as bullet points later on in the Discussion.

11. In the Conclusion section, it could more certainly outline the translational potential of the findings, i.e., how the protein markers determined could facilitate drug development or risk stratification. It now only recapitulates the material of previous sections.

Comments on the Quality of English Language

The English is generally clear and understandable, with good technical terminology and appropriate scientific tone. However, some sentences are overly long and complex, particularly in the Abstract, Introduction, and Discussion, which affects readability.

Author Response

Comments 1: It is relevant and innovative combining plasma proteomics with bidirectional MR for CIHD. Nevertheless, the authors should more directly point out the innovation of their study in contrast with past proteomic GWAS integration studies (e.g. Ferkingstad et al., Nature Genetics 2021) in the Introduction.

Response 1:We have accepted this suggestion and added the following content at the end of the introduction section: “Compared with previous integrative studies, our approach yielded a more robust causal inference by excluding weak instrumental variables and addressing potential reverse causality through bidirectional Mendelian randomization. Furthermore, we identified more promising therapeutic targets for the precision treatment of CIHD via systematic drugability assessment and target prioritization.”

Comments 2:The Abstract is fairly compact. Think about making some sentences shorter and quantifying important findings, for example, number of proteins investigated, number of markers detected, and significant pathways, so the abstract is more readable for the wider audience.

Response 2:After reading this opinion, it was found that there were some issues in the abstract section. We have made some simplifications and modifications to make this part more readable.

Comments 3:Graphics, such as the Manhattan plot, the forest plot, and the KEGG barplot, are indispensable; however, the legends are not detailed enough. It is important to include exhaustive descriptions of the axes, thresholds of significance, and key color codes. Also, Figures 2-5 would benefit from higher resolution as well as clearer presentation for clearer reading.

Response 3:Thank you for your suggestion. We believe that this is necessary. We have replaced some of the clearer pictures and also made numerous additions and modifications to the annotations for the pictures.

Comments 4:Although the flowchart is useful, it could also be improved by the addition of data sources, sample numbers, and crucial analytical protocols, for example, the number of proteins screened at each stage, in order to help the reader negotiate the complex workflow.

Response 4:Thank you for your reasonable suggestions. We have made appropriate modifications to the process roadmap.

Comments 5:It details the MR methods, but the presentation order got mixed up. It might help to reorder according to the actual pipeline: choosing the instruments, forward MR, reverse MR, and testing for sensitivities. It will also help to provide a schematic summary table for the MR methods and assumptions.

Response 5:We agree with the content mentioned in the suggestion. Based on this, we have split and adjusted the "2.3. Bidirectional MR analysis and screening" section.

Comments 6:Explain how multiple testing correction was applied during the course of the MR analyses (e.g., Bonferroni, FDR). It's relevant given the number of proteins screened.

Response 6:It is necessary to perform multiple testing correction. We have added FDR correction in Section 2.2 of the Materials and Methods.

Comments 7:The Discussion cursorily states that the data set consists of European cases only. That is a severe limitation. Authors should elaborate on possible ethnic variation in protein QTLs for CIHD and genetics and how future validation in non-European populations could rectify it.

Response 7:Thank you for your suggestion. We did not take these shortcomings into consideration. In this section, we have added the following contents:For example, the concentration of HP protein-related inflammatory complexes in South Asian males is significantly lower than that in white males (median: 2.5 vs 3.2 ng/L, P < 0.001)[51]. The difference may stem from genetic background (such as linkage disequilibrium patterns) and environmental factors (such as diet, metabolic habits) that are specific to different races[52] .These findings require further validation in Asian and African populations: On one hand, validate the consistency of the causal chain of pQTLs - proteins - CIHD across different ethnic groups using large - sample plasma proteome and GWAS data;on the other hand, optimize the ethnic adaptability of biomarkers by integrating ethnic - specific genetic variations, ultimately enhancing the global universality of the research findings.

Comments 8:The Discussion nicely connects metabolic as well as inflammatory pathways, but the explanation at times is repetitive, particularly for ADH5 and ADH6 in metabolism. Think about simplification in order to highlight the core metabolism–inflammation–regulating axis as well as how it guides therapeutic targeting.

Response 8:We express our sincere gratitude for your valuable feedback. In response, we have carefully deleted and revised the redundant parts in the discussion.

Comments 9:Comprehensive druggability analysis. Nevertheless, criteria for prioritizing therapeutic targets for treatment (CXCL12, PLAU, CD14) might be specified more clearly. It would greatly assist the ranking of candidate targets by evidence strength, involvement in pathway(s), and druggability using a table summarizing the ranking.

Response 9:We believe that it is necessary to take such actions. Accordingly, we have incorporated relevant content regarding the scoring criteria or ranking in the new manuscript and the accompanying appendices.

Comments 10:Their shortcomings are listed, but could have been more organized as well as complete. For example:Confounding by comorbidities, Single ancestry limitation, Potential pQTL detection bias, and Lack of experimental validation. It would make them more readable to list them as bullet points later on in the Discussion.

Response 10:Your suggestions are entirely valid and meaningful. Therefore, we have added "Table A2. Summary of research limitations and corresponding improvement strategies." to the appendix.

Comments 11:In the Conclusion section, it could more certainly outline the translational potential of the findings, i.e., how the protein markers determined could facilitate drug development or risk stratification. It now only recapitulates the material of previous sections.

Response 11:After reading your suggestions, we have realized that there are indeed some deficiencies here. So we made appropriate additions and modifications to the Conclusion section.

Overall, I am extremely grateful for your valuable suggestions. Your rigorous attitude and conscientious spirit in academic work are qualities that I can learn from. The newly uploaded manuscript has undergone multiple revisions, but there are still some issues. You can read the new version again. I look forward to receiving your further comments.

Round 2

Reviewer 3 Report

Comments and Suggestions for Authors

The resubmitted manuscript has been significantly improved. The authors properly and effectively responded to all prior comments. The Introduction clearly indicates the innovativeness of the study, and the Methods are more comprehensive with specific datasets and FDR correction. Figures and legends are better, and the Discussion briefly describes the metabolism-inflammation axis and translational significance.